# Optimization System Based on Convolutional Neural Network and Internet of Medical Things for Early Diagnosis of Lung Cancer

**DOI:** 10.3390/bioengineering10030320

**Published:** 2023-03-02

**Authors:** Yossra Hussain Ali, Varghese Sabu Chooralil, Karthikeyan Balasubramanian, Rajasekhar Reddy Manyam, Sekar Kidambi Raju, Ahmed T. Sadiq, Alaa K. Farhan

**Affiliations:** 1Department of Computer Sciences, University of Technology, Bagdad 110066, Iraq; 2Department of Computer Sciences & Engineering, Rajagiri School of Engineering & Technology, Kochi 682039, Kerala, India; 3School of Computing, SASTRA Deemed University, Thanjavur 613401, Tamil Nadu, India; 4Amrita School of Computing, Amrita Vishwa Vidyapeetham, Amaravati Campus, Amaravati 522503, Andhra Pradesh, India

**Keywords:** lung cancer detection, deep learning, internet of medical things, convolutional neural networks, and particle swarm optimization

## Abstract

Recently, deep learning and the Internet of Things (IoT) have been widely used in the healthcare monitoring system for decision making. Disease prediction is one of the emerging applications in current practices. In the method described in this paper, lung cancer prediction is implemented using deep learning and IoT, which is a challenging task in computer-aided diagnosis (CAD). Because lung cancer is a dangerous medical disease that must be identified at a higher detection rate, disease-related information is obtained from IoT medical devices and transmitted to the server. The medical data are then processed and classified into two categories, benign and malignant, using a multi-layer CNN (ML-CNN) model. In addition, a particle swarm optimization method is used to improve the learning ability (loss and accuracy). This step uses medical data (CT scan and sensor information) based on the Internet of Medical Things (IoMT). For this purpose, sensor information and image information from IoMT devices and sensors are gathered, and then classification actions are taken. The performance of the proposed technique is compared with well-known existing methods, such as the Support Vector Machine (SVM), probabilistic neural network (PNN), and conventional CNN, in terms of accuracy, precision, sensitivity, specificity, F-score, and computation time. For this purpose, two lung datasets were tested to evaluate the performance: Lung Image Database Consortium (LIDC) and Linear Imaging and Self-Scanning Sensor (LISS) datasets. Compared to alternative methods, the trial outcomes showed that the suggested technique has the potential to help the radiologist make an accurate and efficient early lung cancer diagnosis. The performance of the proposed ML-CNN was analyzed using Python, where the accuracy (2.5–10.5%) was high when compared to the number of instances, precision (2.3–9.5%) was high when compared to the number of instances, sensitivity (2.4–12.5%) was high when compared to several instances, the F-score (2–30%) was high when compared to the number of cases, the error rate (0.7–11.5%) was low compared to the number of cases, and the computation time (170 ms to 400 ms) was low compared to how many cases were computed for the proposed work, including previous known methods. The proposed ML-CNN architecture shows that this technique outperforms previous works.

## 1. Introduction

The development of Internet of Things (IoT)-based medical services and sensor systems has created numerous new study topics. IoT, cloud services, and intelligent systems (SIs) have all been covered by the current literature, changing traditional health services into intelligent healthcare. Health practitioners can enhance their practice by using essential technologies such as artificial intelligence (AI) and SIs [1]. The confluence of IoT and AI in the healthcare industry provides a wide variety of options. In light of this, the present study proposes a new AI- and IoT-based illness diagnostic paradigm for intelligent health systems. Due to the enormous cost of health insurance and the prevalence of multiple diseases, the medical center must be converted to a user system as soon as possible [2]. Researchers discovered that the survival rate is much higher when cancer is detected early. However, the early detection of such conditions is a difficult task. According to the correct classification, around half of cancer patients are in the middle or late stages of the disease [3]. In general, medical-image-processing-based disease diagnosis follows the simple procedure described below:Image Preprocessing: In this step, image quality is improved by removing undesirable distortions in the image and correcting the geometric transformation of the image (rotation, scaling, and translation). Some of the essential preprocessing tasks are as follows: denoising, contrast enhancement, normalization, and so on [4].Feature Extraction: This step converts the given input data into a set of features in a process known as feature extraction. In medical image processing, feature extraction starts with the initial set of reliable data, and feature extraction is performed on borrowed values known as features [5]. This makes the classification method much more satisfactory for prediction. Most medical image classifiers are based on the result of feature extraction in CAD.Feature Selection: This step reduces the feature space extracted from the whole image, enhancing the detection rate and reducing the execution and response time. The reduced feature space is obtained by neglecting the irrelevant, noisy, and redundant features and choosing the feature subset that can achieve good performance with the subject of all metrics. In this stage, the dimensionality is reduced [6].Classification: In this stage, the selected features are applied to classify each sample to denote the desired class. However, classification is a big issue in medical image analysis tasks. For automatic classification, optimal sets of features are used, and this step needs to be further improved for the better clinical analysis of the human body for diseases [7]. Computer-aided diagnostic systems have proven efficient in finding and diagnosing problems more rapidly and accurately. Unfortunately, this diagnostic technique is prone to several flaws and ambiguities, which might lead to an inaccurate prognosis. Hence, IoT sensor information is fused into the classification model. Machine learning (ML) models are currently used to indicate lung disease in computerized tomography (CT) images. The pipeline architecture is given in Figure 1. The following are the paper’s significant contributions [8]:Improved CNN Model: The CNN’s efficiency is optimized to boost the lung cancer diagnostic performance, and its computation is based on various inputs. The CNN calculates the improved range of values based on the extracted features [9].Hyperparameter Tuning: The learning parameters for the CNN must be generated on demand. It is also regarded as an optimization problem in which hyperparameters are tuned using the particle swarm optimization (PSO) algorithm [10,11,12].Significant Performance Gains of Diagnosis Complexity: The optimized CNN architecture employs fewer convolutional layers by considering relevant information from the convolutional, fully connected, and max-pool layers. The proposed optimized CNN can reach minimal complexity as a result of the primary structural view of the CNN [13,14].Consideration of IoT Sensor and Medical Images: A combination of sensor information and medical images is used to diagnose lung cancer from pictures [15,16].

Figure 1 shows the pipeline for IoMT-based lung disease diagnosis together with the design of the research model. The scientific effort has been given a defined road map.

### Motivation for Lung Cancer Research

People affected by the disease daily come from many regions of the country and lose their identity, pleasures, and gifted quality of life. These facts inspired us to identify and analyze the disease’s underlying causes and address the need for the early disease detection of lung cancer. This motivated us to research lung cancer as soon as possible. Other factors are summarized below:

(i) Earlier diagnosis can improve the patient’s survival and quality of life.

(ii) For people who have been diagnosed with lung cancer, research can provide a better and longer future.

(iii) It can also, in the end, improve the number of survivors.

In advanced medical images, the internal organs and tissues of the body are seen using one or more specialized diagnostic techniques, referred to as “advanced imaging.” Without any human oversight, the CNN automatically finds essential characteristics. A multi-layer perceptron that can automatically learn the distribution law of data from a massive dataset has been created using the convolutional neural network (CNN).

The convolutional neural network (CNN) uses a significant quantity of data to identify the complicated representation of visual input. Because humans can perceive colors, it is essentially inspired by the human visual system. Convolutional networks recognize images as volumes, which represent three-dimensional objects. Red, Green, and Blue (RGB) encoding and the mixing of these three colors create the color spectrum in digital pictures. Each layer produces a dot product of the input pixels before moving on to the following layer. Along with the fundamental CNN structure, the CNN contains the following levels of activities.

In particle swarm optimization (PSO), the potential solutions are particles. At the optimal level, particles traverse the problem space and are linked with fitness, which is the best option. Each particle also maintains a tracking of all of its coordinates. Pbest is the new name for this fitness value. A particle’s value is the best overall value and is referred to as the best if it accepts the whole population as its neighbor.

The elements of the study project are (2) primary knowledge, which aids the reader in understanding the fundamental idea; (3) related works, which piece together the main principles; (4) system model, intended for the essential features; (5) experimental results and discussion, where the results are extensively discussed; (6) the conclusion, which discusses the results, viewpoints, and references for a simple comprehension of the work.

The rest of the paper is structured as follows: Section 2 discusses the preliminary knowledge about lung cancer diagnostic methods. Section 3 describes the related work, which elaborates on the current state-of-the-art methods for the classification of lung cancer. Section 4 discusses the proposed system model for lung cancer diagnosis. Section 5 discusses the experimental results for the proposed as well as existing methods. Section 6 concludes the paper and also explains future work.

## 2. Primary Knowledge

Current classification methods can be divided into two categories: supervised or unsupervised classification methods. In SVM, the Gaussian kernel is used; when there is no prior knowledge of the data, it is utilized to execute the transformation. Random Forest is used throughout the cited studies. Because of its efficiency and short processing time, SVM is the best technique for finding the best model and has received growing interest. An SVM algorithm for lung cancer disease does not support small-scale datasets and also requires a long processing time [17].

The automatic segmentation of components present inside an image, such as the region of interest in diagnostic images, has improved because of new methodologies combining transfer learning and techniques such as fine-tuning [18]. Deep learning and artificially intelligent techniques are applied to this data collection to develop statistics. Professionals then analyze the data to help customers lead healthier lives and avoid illness. Patients’ data are collected via health records (HRs), IoT sensor systems, personal and mobile phones, internet data, and social networks, which are emphasized among recent innovations. The PH uses artificial intelligence (AI) approaches to enhance illness development, illness prognosis, self-regulation and the self, and clinical intervention procedures using the set of data generated [19]. Recent studies are progressively looking into IoT devices in the healthcare profession. Healthcare diagnosis is routinely aided. Lung cancer patients will have a better chance of surviving if they can be diagnosed early. Deep learning and machine learning are frequently employed in the screening and effective identification of lung disease. This study’s major goal is to examine the function of deep learning in identifying and diagnosing lung disease. As a result, we employed a convolutional neural network (CNN), a type of deep learning model, to diagnose lung disease using radiography imaging [20]. A CT scan of the lungs can be used to identify and arrange the lung spikes and determine how dangerous they are. An improved CNN has high accuracy and temporal complexity characteristics; however, this is not as responsive to the setup as prior frameworks. They have tumors.

## 3. Related Work

A survey of roughly 36 articles that employed computational methods [21] to forecast various diseases was carried out in this study. The research consists of a variety of neural network models that are used to identify a range of illnesses and identify gaps that can be filled in the future for identifying lung disease in healthcare IoT. Every method was examined stepwise, and the general disadvantages were identified. Furthermore, the review examined the type of information utilized to predict the disease in question, that is, whether they were benchmark or manually obtained data. Lastly, strategies are defined and shown based on several accessible approaches. This will aid research in the future in correctly identifying malignant patients in their initial stages with no defects.

In [22], the authors propose an efficient SVM for lung classification, wherein the SVM’s characteristics are adjusted, and features are extracted using a refined gray wolf optimization algorithm in combination with a genetic algorithm (GWO-GA). The experimental phase is classified into three categories: parameterization testing, feature engineering, and optimal SVM. A benchmarking image dataset of 50 low-dose and archived lung CT scan images was used to monitor the effectiveness of the provided technique. The given work demonstrated the performance on the entire sample image dataset in various ways. Furthermore, it has an accuracy rate of 93.54%, which is much better than the evaluated approaches.

The current study presents a novel prototype for the feature extraction and classification of computed pulmonary tomography based on the Internet of Things. Combining the backpropagation algorithm with Parzen’s probability density, the presented approach employs a Software Development Kit (API) known as the Internet of Medical Things to identify lung images. The method worked well, with more than 98% correct classifications of lung images. The model then moves on to the lung segmentation method, which creates a lung map and uses fine-tuning to detect the pulmonary boundaries on the CT image using the Masks R-CNN network. The hypothesis was proven correct, as the suggested approach outperformed previous efforts in the research, achieving categorization parameters such as 98.34% [23].

Again, for evaluation, this work utilized computed tomography (CT) images of the lung and a probabilistic neural network (PNN) for diagnosis. After the raw lung images are preprocessed, features are extracted using the Gray-Level Co-Occurrence Matrix (GLCM). The chaotic crow search algorithm (CCSA)-based selection of features is presented. The calculation metrics employed were precision, sensitivity, predictive value, and accuracy. According to the investigation, on pictures with unevenly distributed cells, the feature selection based on CCSA effectively reaches 90%.When pulmonary development and degeneration are out of proportion, cancer cases of different kinds arise as a result; cancer cells then grow and expand, eventually creating a tumor. Whenever atmospheric poisons contact lung tissue, they cause harm. The unclear features of the lung cancer nodule make computer-aided automated diagnosis difficult at the moment. A selection approach to extract key characteristics was utilized together with the average accuracy to improve the classification performance. According to the results, the PNN with CCSA-based selection features outperforms the PNN without such features [24].

Researchers proposed a novel Internet of Things (IoT)-based predictive model based on fuzzy-cluster-based enhancement and classification to forecast lung cancer development via ongoing analysis and to enhance health by delivering medical instructions. For successful picture segmentation, the fuzzy clustering approach is utilized, which is based on the energy that can be transferred through extraction. In addition, the Fuzzy C-Means Clustering technique is employed to classify the transition area characteristics from the lung cancer image features. This method uses the Otsu thresholding approach to recover the transition zone from a lung image. In addition, the right-hand-side images and the morphology reduction procedure are applied to improve segmentation accuracy [25].

The morphology cleansing and image region filling procedures were conducted across an edge’s lung image to obtain the image areas. The authors also present a novel continuous classification approach that integrates existing Association Rules (ARM), the conventional Decision Tree (DT) with feature representation, and a CNN. LungNet, a unique combination DNN-network-based framework built using CT scans and healthcare IoT data obtained using a wearable camera, is presented to address the issue. LungNet is made up of a 22-layer CNN model that blends latent factors acquired from CT scan pictures with MIoT data to improve the system’s detection ability. This system, which is controlled from a centralized server and was trained using a balanced dataset of 525,000 images, can classify lung disease into five classifications with high precision (96.81%) and a lower false-positive rate (3.35%) than comparable CNN-based classifications. Furthermore, it accurately differentiates phase 1 and phase 2 lung tumors into 1A, 1B, 2A, and 2B sub-classes with a probability of detection of 10.5% [26].

All of the devices in this system are responsive to VOCs linked to lung cancer. The device’s data had to be normalized before they could be used. Next, to improve classification success, characteristics were collected from the data. Principal Component Analysis was performed to eliminate superfluous characteristics. Finally, k-Nearest Neighborhood and SVM algorithms were used to classify the collected characteristics [27].

This research used computed tomography (CT) images to develop automatized learning with quantitative research utilizing lung identification and therapy. On CT samples acquired from a lung dataset, the proposed model applies a Gaussian filtering (GF)-based preprocessing method. Moreover, they are put through the normalized cuts (Nuts) method, which may spot nodules that existed before the image. In addition, the oriented FAST and rotated BRIEF (ORB) technique is used as a feature extractor. Finally, the sunflower optimization-based wavelet neural network (SFO-WNN) model is used for lung cancer diagnosis. A descriptive analysis was conducted to evaluate the diagnosis results of the MLDS-LCDC model, and the findings were tested based on many factors [28].

The use of automated and optimal computer-aided diagnosis for lung disease is proposed in this paper. The procedure begins with the normalization and noise removal of the input images in a preliminary step. After that, Kapur entropy maximization and classification techniques are used to divide pulmonary areas. In the remainder of this study, 19 GLCM features were collected from the segmented images. The images with the highest priority were then chosen to reduce the program’s complexities. The feature selection method is considered a new optimization approach known as Enhanced Thermal Exchange Optimization (ITEO), which aims to increase reliability and stability. ITEO uses an optimized artificial neural network to classify images into healthy or malignant situations [29].

Magnetic resonance imaging (MRI), computed tomography scans (CT scans), ultrasonic devices, and positron emission tomography (PET) have all been used widely in the past; nevertheless, these methods have significant disadvantages, including cost, as well as the existence of radioactive material. Because of its convenience, accessibility, low price, and convenience of use, sensor synchronization in processing has recently been adopted in advanced screening. In terms of medical use, biotechnologies are becoming common tools for cancer scares, and intelligent sensor advancement has been realized. This article describes the results of studies on the use of biosensing in the treatment of lung cancer. According to the study, the electrochemical biosensor is the most widely utilized for early-stage lung cancer diagnosis [30].

In this study, the researchers present a new deep learning approach for lung cancer detection, which combines a convolutional neural network (CNN) with kernel k-means clustering. The proposed methodology was tested using the Anti-PD-1 Immunotherapy Lung datasets from The Medical Image Archives (https://www.kaggle.com/datasets/nih-chest-xrays/data and https://comport.com/healthcare-it-solutions/medical-image-archives/, accessed on 19 October 2022). The researchers utilized 400 MRI images from these data, which were carefully classified and included 150 routine lung scans and 250 cancer images. The framework was used to analyze the information throughout the first step. The kernel k-means clustering approach flattens neurons in the feature space for each image resulting from the convolution operation in the CNN. The center of each cluster is again determined using this procedure, which provides the prediction classification of each piece of data in the cross-validation. Various data points were used in k-fold cross-validation to assess the effectiveness of the suggested approach. When employing kernel functions with sigma = 0.05 in nine-fold cross-validation, the presented approach achieved the best set of metrics, with 98.85%, 98.32% sensitivity, 99.40% precision, 99.39% specificity, and a 98.86% F1-measure [31].

Employing expanded SegNet and convolutional neural network models, an intense CAD system for cancer detection with breast tomography images was constructed in this study (CNNs). To segment the lung from chest CT images, a dilated SegNet model was used, and a CNN model with batch normalization was built to identify the real nodules among all probable nodules. Using example cases from the LUNA16 dataset (https://luna16.grand-challenge.org/, accessed on 19 October 2022), the expanded SegNet and CNN classifiers were trained. The nodule classification was assessed using sensitivity, and the segmented model’s efficiency was quantified by the Dice coefficient. The statistical method confirmed the detection accuracy of the features obtained by CNN classification [32].

Another study provided a combined lung lesion detection and recognition network that can simultaneously identify, segment, and classify pulmonary nodules while accounting for inherent labeling uncertainties in the training dataset. It allows the final identification of nodules and the segmentation of the discovered nodules. For the improved analysis of 3D data, both the nodule identification and categorization subnets of the proposed combined system use a 3D encoder–decoder design. Furthermore, the categorization subnet boosts the categorization model by implementing characteristics extracted from the detection subnetwork with multiresolution nodule-specific data. When compared to the current backpropagation method, the proposed method provides useful prior knowledge to better optimize the much more difficult 3D classifier correlational design and better identify worrisome clusters from several other tissues [33].

Integrating two feedforward attentiveness processes, one at the nodule level and the other at the item level, a hierarchy attention-based multiple instance learning (HA-MIL) paradigm for clients diagnosing the disease was developed. The suggested HA-MIL platform was built by combining important parts’ representations with nodule information and then combining the nodule representation with lung recognition. The HA-MIL framework outperformed existing techniques, such as higher transaction learning in the case of MIL and embedding-space MIL, on the general populace Lung Image Database Consortium and Set Of images Resource Initiative (LIDC-IDRI) dataset (https://paperswithcode.com/dataset/lidc-idri, accessed on 19 October 2022), demonstrating the efficacy of hierarchy ensemble learning. The HA-MIL model also detected the important nodules and qualities by using greater attentiveness weights, according to the analysis of the findings [34].

Traditional Chinese Medicine (TCM) has been proven to be an excellent therapy for treating lung disease, and precise syndrome classification is critical to treatment. The creation of smart TCM syndrome discrimination algorithms is enabled by concrete evidence of TCM treatment instances and the advancement of intelligent systems. This is intended to broaden the scope of TCM’s advantages for lung cancers. This study aimed to develop end-to-end TCM diagnostic systems that might mimic lung disease classification. The developed models employed unorganized patient files as inputs to take advantage of data gathered by lung professionals for actual TCM therapy instances. The resulting algorithms outperformed those that used unstructured TCM data [35,36].Even though the articles in the related work have their significance, they can still increase their accuracy and performance. Through the understanding gained by the analysis of the above articles, this research has identified that ML-CNN with PSO and hyperparameters provides higher accuracy and greater performance and precision:SVM with wolf optimization and the genetic algorithm has 93.54% accuracy;R-CNN with the backpropagation algorithm and Parzen’s probability density has 98.34% accuracy;The Gray-Level Co-Occurrence Matrix and chaotic crow search algorithm with a probabilistic neural network provide 90% accuracy;The CNN with Association Rules and conventional Decision Tree (DT) provides 96.81% accuracy;Gaussian filtering with the sunflower optimization-based wavelet neural network has less than 90% accuracy;Enhanced Thermal Exchange Optimization and Kapur entropy maximization and classification techniques have less than 90% accuracy;The convolutional neural network, in nine-fold cross-validation, provides 98.32% accuracy.

However, our approach, ML-CNN with PSO and hyperparameters, provides 98.85% accuracy, which is the highest among all compared methodologies.

## 4. System Model

In this section, the proposed system model is given in detail. Through IoMT technologies, sensors, and devices, the classification of lung cancer is implemented. Figure 2 illustrates the proposed model in detail.

### 4.1. Preprocessing

Firstly, lung images are denoised to improve the quality of the images. In this work, the Bayesian threshold-based Taylor series approach is proposed for denoising noisy pixels into better-quality pixels. This proposed approach is based on the Taylor series since it provides high-resolution wavelet subbands. The Taylor series defines an infinite sum of terms that are estimated from derivative functions in which the values are pixels and are varied. As a result of this, the Taylor series for a 3D frame is expressed as
(1)$$T\;\left(X\right)=\left(a\right)+{\left(X-a\right)}^{T}\;Df\left(a\right)+\frac{1}{2!}\;{\left(X-a\right)}^{T}\;\;\left\{D^{2}\;f\left(a\right)\right\}\left(X-a\right)+\ldots$$

Figure 2 clearly illustrates the system architecture and functions for core comprehension. Figure 2 explains the ML-CNN and feature extraction models together with all preprocessing concepts.

In general, an image comprises different pixel values since the intensities of an image are not similar in all places of an image. Therefore, the changes in pixel values need more attention for denoising. From the defined Taylor series, the noise in subbands is computed and eliminated if it is greater than the estimated Bayesian threshold. The threshold for all subbands is formulated using the following expression:(2)$$T_{B}=\frac{\sigma_{N}^{2}}{\sigma_{s}}$$
where ***σ_N_*** represents the noise estimation, and ***T_B_*** is the threshold obtained for all bands.

When $\sigma_{s}\neq 0$, it is formulated as
(3)$$\sigma_{s}=\sqrt{max\left(\left(\sigma_{y}^{2}-\sigma_{y}^{2}\right),0\right)}$$
where ***σ_y_*** is defined as
(4)$$\sigma_{y}=\frac{1}{N}\;\left(\sum SB_{i}\right)$$

The subbands *SB_i_* are $SB_{i}=\left\{\{LLL,\;LLH,\;LHL,\;LHH,\;HLL,\;HLH,\;HHL,\;HHH\right\}$, which is considered the total number of subbands, and *σ_N_* is determined by
(5)$$\sigma_{N}=\left[\frac{median\;\left(SB\right)}{0.6745}\right]$$

Upon estimating the ***T_B_*** value, the obtained values of the Bayesian threshold are arranged, and then using the curve fitting method, an expression is written as follows:(6)$$ϒ=\frac{p_{1\;}\;\beta^{2}+p_{2\;}\;\beta +p_{3}}{\beta +q}$$

In Equation (6), β defines the standard noise deviation, and the values of other terms are given as ***p*_1_ = 0.9592**, ***p*_2_ = 3.648**, ***p*_3_ = −0.138**, and q = 0.1245. Next, a BM3D filtering algorithm is applied, and its mathematical formulation is expressed as follows:(7)$$\hat{Y}S_{SB}=T_{3D}^{-1}\;\left(\propto \left(T_{3D}\left(Z_{SB}\right),\;T_{B}ϒ\sqrt{2\;log\left(N^{2}\right)\;}\right)\right)$$

In Equation (7), ***T*_3*D*_** is the unitary 3D transform, $\hat{Y}S_{SB}$ is denoted as stacked subbands, Z is the subband size, and ∝ defines the threshold operator. From this final formulation, each subband is reconstructed and attains denoised frames for feature extraction.

Equation (12) $D_{RE}\;\left(F_{i},F_{i+1}\right)$ denotes the distance measure of Relative Entropy, and $D_{RE}\;\left(F_{i},F_{i+1}\right)$ gives the distance measure and Square Root of Relative Entropy between the frames. The term $P_{i}=\left\{P_{i}\;\left(1\right),\;P_{i}\;\left(2\right)\ldots \;P_{i}\left(n\right)\right\}$ represents the probability distribution function of frames present in a video, i.e., given as Fi and Fi+1, obtained from a normalized intensity histogram of n bins. The value of the bin is n = 256, and k represents the total number of frames present in each shot. Upon estimating the distance measures based on Relative Entropy and the Square Root of Relative Entropy, a comparison is made between them based on the time direction. From this comparison, the alterations in frames are identified, and redundantly occurring frames are eliminated. In this step, image intensity values are normalized into equal ranges for correct processing at the next level. The next step is executed as an intensity normalization process, in which the range is between 0 and 1. The Z-score normalization function is used for intensity normalization, the performance of which is better than the decimal and min–max normalization techniques. This normalization function is computed as follows:(8)$$Y_{i}=\frac{x_{i}-\mu }{\delta }$$
where ***Y_i_*** represents the normalized intensity values for the input image, μ is the mean intensity value, δ represents the variance, and x_i_ represents the intensity position. The Z-score normalization function was used for intensity normalization, which was carried out in the processing of the image for upcoming processes, in the rational range (0–1). This step is mandatory for all images for both training and testing operations. This minimizes the overhead or complexity for similar medical image retrieval processes with the same properties (size, intensity values, and so on).

### 4.2. Feature Extraction

Table 1 shows the list of extracted features using the above-mentioned methods. The following features were re-extracted from the lung images:Intensity-based features: These types of features describe the information on the intensity and histogram values located in the image (visibility is considered in the grayscale or color intensity histogram).Shape-based features: These give image information about the shape and size.Texture-based features: These describe the image intensity variations on the surface and estimate the properties of coarseness, smoothness, and regularity.

After feature extraction, the optimal set of features is extracted since the classification method cannot process a huge set of features. For this reason, the best sets of features are selected from the extracted feature set by spatial transformer networks.

The CNN is one of the types of neural network schemes with a major problem, i.e., high training complexity. Training depends upon loss values; training is frequently required. To address this issue, in this paper, the PSO algorithm is proposed, which finds the optimal values of the hyperparameters according to the input types and ranges. This type of integration helps reduce the hardware cost for CNN training and further produces fewer epochs for training and testing with the use of epochs. Further, this combination reduces the local minimum issue with regular backpropagation methodology training. The structure of the CNN is represented as follows:Convolution Layers: This is the input layer that acquires the inputs, i.e., signals. This layer normalizes the range of all inputs into a single value for fast processing.Pooling Layers: This layer’s purpose is to minimize the input size, which automatically influences the higher performance.Activation Layer: This layer uses the ReLU activation function, which maximizes the outputs for non-linear cases.Fully Connected Layer: This layer uses an activation function called softmax to control the output range. The proposed CNN computes the weight values for a given problem between neighboring neurons.

In Figure 3, deep neural networks called convolutional neural networks (CNNs) are shown classifying and segmenting pictures. Both supervised and unsupervised machine learning techniques can be used to train the CNN. With no loss of information, its built-in convolutional layer reduces the high dimensionality of pictures. In Table 2, low-level extracted features are listed, which are used for fundamental characteristics automatically retrieved from a picture without needing shape information or focusing on proximity. Table 3 shows lung nodules that are very typical and typically not a reason for alarm. Even so, finding out that one has a lesion on the lung might be scary.

Figure 4 describes the flowchart for the proposed optimized CNN with the PSO algorithm. The fitness function is computed as follows:(9)$$fitness_{function}=1-\frac{1}{N}\sum_{i=1}^{N}{\left(Y_{i}-{\hat{Y}}_{i}\right)}^{2}$$

Based on the fitness function, the PSO algorithm computes the position of the neighboring particle that is influenced by the current position, and hence, each particle is updated using
(10)$$V_{i}\left(t+1\right)=\left(c_{1}\times RAND\;()\times \left(p_{i}^{best}-p_{i}\left(t\right)\right)\right)+\left(c_{2}\times RAND\;()\times \left(p_{g}^{best}-p_{i}\left(t\right)\right)\right)+V_{i}\left(t\right)$$
(11)$$p_{i}\left(t+1\right)=p_{i}\left(t\right)+v_{i}\left(t\right)$$

If particle $p_{i}\left(t\right)$ lies close to the local best $p_{i}^{best}$**,** the second term in the velocity-updating rule approaches zero, and for particles $p_{i}\left(t\right)$ lying close to their global best $p_{g}^{best}$, the third term approaches zero. So, the particles close to their local best $p_{i}\left(t\right)$ or global best $p_{g}^{best}$ can evolve more easily than those far away from their best ones. The velocity-updating rule is as follows.

For the overall CNN layers, the performance of parameters used in training and testing is tuned using the PSO algorithm. With the use of global and local search issues, the performance is optimized. The hybrid CNN with the PSO is described below.

$V_{i}\left(t+1\right)$ represents the ith particle’s velocity, which is updated for every iteration. Other variables are the particle’s best position and coefficient variables. In the softmax layer of the proposed ML-CNN model, entropy functions are used for disease diagnosis and the threshold update to the given set of inputs. For the classification into the appropriate classes, the features are extracted, the similarity between probability distributions is calculated using Relative Entropy, and an additional entity is measured for determining the threshold values for the classifications, which is estimated using the Square Root of Relative Entropy. In this work, the features are extracted in the left direction toward the right for the accurate classification of the class. For this approach, the values of Relative Entropy and the Square Root of Relative Entropy are determined from the following formulations. In Algorithm 1 machine learning CNN classification pseudocode is presented for clear understanding purpose.
**Algorithm** **1.** ML CNN classification pseudocode*Input: Sensed and image patterns F_fn_**Let C    be the size of the total sequence to be classified**1. N_i_ = 3* *// no. of layers**2. N_r_ = 1*  *// no. of runs**3. Let N_s_ // no. of samples**4. Let N_f_ // no. of features**Initiallize traning set size (t _фc_)**Testing set size (t _ψc_)**// 80% training and 20% testing**Extract features from the sequence and create a list of features**Let ffn be the feature set extracted**Initialize CNN parameters**5. Let batch size = 1**6. No. of epochs = 1**Let L_bs_ be the labels corresponding to the selected features**Let N_c_ be the number of classes to be identified**7. Load f_fn_*  *// load optimized data**8. For i=1 to N_l_*   *Split F_fs_ (feature set into T (feature subset))**9.*   *For j = 1 to T**To find backpropagation CNN**10.*     *For i = 1 to size (t _фc_)**11.*       *N_1_. F_fn_ = t _фc_ (i,:)**12.*     *End**13.*     *For i = 2:N_l_**14.*      *Layer = i**15.*      *if N_l_(i) = t _фc_ (i,:)**16.*      *Val = N_l_(i) ∗ t _фc_ (i,:)**17.*         *For j = 1:length(Val)**18.*            *z = 0;**19.*            *For k = e:length(Val)**20.*              *Kk = Kk + 1**21.*              *Val = N_l_(i-1) ∗ t _ψc_ (k)**22.*              *Val1 = N_l_(i) ∗ t _ψc_ (k(;,:,1)**23.*             *End**24.*            *End**25.*          *End*    *Let f_fn_ be feature in F_fs_(i,j)*    *Trained TCLF estimates f_fn_**26. Let R_sort_= sort(T_out_)    //rank level**27. Let accuracy = mean(T_clf_,1)**28. T_clf_ = ∑_i_(T_out_/R_sort_)**29.*      *cnti=∑ (F_fn_) in belonging to samples N_s_**30.*     *End**31. Compute Total count as Cnt_T_ = *$\sum_{i}^{n}\;\;$*Cnt_i_*   *Compute probabilistic components for each class as**32.*       *For i = 1 to N_c_**33.*       *Pcomp(i) = cnt_i_/cnt_T_**34.*      *End**35.*  *End (Ending of firt i for loop)**Output: Classified output*
(12)$$\left(F_{i},F_{i+1}\right)=\sum_{k=1}^{n}P_{i}\;\left(k\right)log\frac{P_{i}\;\left(k\right)}{P_{i+1}\;\left(k\right)}\;$$
(13)$$D_{SRRE}\;\left(F_{i},F_{i+1}\right)=\sqrt{\sum_{k=1}^{n}P_{i}\;\left(k\right)log\frac{P_{i}\;\left(k\right)}{P_{i+1}\;\left(k\right)}}$$

## 5. Experimental Results and Discussion

### 5.1. Implementation

This study involved creating a fast and smart lung cancer detection system for use in the healthcare context. It classifies individuals with lung concerns using a deep-learning-based approach on a lung database. The information on the characteristics of the lung was determined using the ML-CNN approach. These data are used to guide the search area for the ML-CNN technique to determine the suitable class again for provided inputs using the PSO algorithm. The entire study was carried out using the Python programming language. The results are reported and organized into tables and graphs for comparison to other assessment parameters. The necessary description is provided in Table 4 and the CNN is learning using hyperparameters and PSO algorithms, which minimize iteration and time.

The above settings are applied to prevent overfitting. The hyperparameters are calculated using shallow tree and deep tree models. Once we do not have enough data, we can use the four-fold validation method; as such, it is not a heuristic method.

The above values are called hyperparameters and define the training required to build models. Changing the values of the hyperparameters changes the generated model and is generally used to determine how the algorithm learns relationships in the data.

A set of examples that are not part of the training process and do not participate in model tuning are used to evaluate the test set’s performance. This toolkit does not leak data into models, so it can be safely used to get an idea of how the model will perform in production.

During implementation, various hierarchical models of the PSO algorithm were adjusted. For such systematic methods, a variety of performance measures were employed, all of which were generated from the CNN model. The performance indicators that we utilized in our research are listed below.

### 5.2. Dataset Description

(i) LISS Dataset (A Public Database of Common Imaging Signs of Lung Diseases) (date of access: 19.10.2022) 

We abbreviate this dataset as LISS CISL‖, a public database for researchers and academic purposes. It contains 511 2D CISLs for 252 patients, 166 3D images for 19 patients, and nearly 9 lung nodule signs captured at the Cancer Institute and Hospital at the Chinese Academy of Medical Sciences. These images were captured by GELightSpeed VCT-64 and Toshiba Aquilion 64 Slice CT scanners, Holt, MI, USA. The image format is DICOM 3.0. In the LISS database, 2D instances are covered by the nine categories of lung nodule sign images, and 3D instances are covered by only one sign (ground-glass opacity (GGO)).

In LISS, all CT images are assigned their corresponding label, and each image size is in pixels. The main reason why we considered this database is that it comprises a large number of lung images with different nodule signs. The lung image slice size in each class is 0.418–1 mm, and the mean value of the slice is 0.664 mm. The LISS database description is depicted in Table 5, which is categorized into two classes: CT imaging scans and annotated ROIs.

In the LISS database, the CT imaging scans are stored in plain text, and all of the annotated ROI CT images are stored as an individual file per nodule sign. The description of each nodule sign is given below:GGO: This is a sign showing hazy maximized attenuation in the lung and vascular and bronchial margins. It is suggestive of lung adenocarcinoma and bronchioloalveolar carcinoma.Lobulation: This is an indication that the septae of connective tissue, which contain fibroblasts recognised as coming from the perithymic mesenchyme, are growing (malignant lesion).Calcification: This is a sign of the deposition of insoluble salts (magnesium and calcium). Its characteristics, such as distribution and morphology, are significant in distinguishing whether a lung nodule is benign or malignant. Popcorn and dense calcifications indicate that the given lung image is benign, and the central regions of lesions, spotted lesions, and irregular appearance indicate malignancy. It is determined from lung nodule areas with high-density pixels, and CT scan ranges are above 100 HU (Hounsfield Units).Cavity and Vacuoles: These are signs of hollow spaces, represented by tissues. The vacuole is a type of cavity. It is suggestive of bronchioloalveolar carcinoma and adenocarcinoma, and the cavity is connected to tumors greater than 3 mm.Speculation: This is a sign caused by stellate distortion in tissue lesions. The cancerous tumor’s intrusion causes this, which is one of the common lung nodule signs correlated with a desmoplastic response. This result indicates the fibrotic strand radiates with lung parenchyma. It is a sign of a malignant lesion.Pleural Indentation: This is a sign of a tumor-affected area, which is diagnosed in the tissues. The sign is connected with peripheral adenocarcinomas that comprise central or subpleural anthracitic and fibrotic foci.Bronchial Mucus Plugs (BMPs): These are also referred to as focal opacities. They vary in density and may have a liquefied density, which is above 100 HU. They are due to allergic bronchopulmonary aspergillosis and represent Intrabronchial Air, which the mucus transforms.Air Bronchogram: This is a sign of the formation of low-attenuation bronchi over the background of the high-attenuation airless lung. It is due to (1) proximal airway patency, (b) the removal of alveolar air, which produces absorption (atelectasis) or replacement (pneumonia), or (3) the presence of both symptoms. In very rare cases, the air displacement is the result of the marked Interstitial Expansion (Lymphoma).Obstructive Pneumonia: This rectangle mark available in Figure 5 is a sign of a small lung nodule volume, which is due to distal collapse. This is mainly due to proximal bronchial blocking. It is combined with adenocarcinoma and squamous cell carcinoma. For a visual representation, it represents the cuneate or flabellate area with increased density. Among the descriptions of lung nodule signs, radiologists have concluded that the GGO, Cavity and Vacuoles, Lobulation, Bronchial Mucus Plugs, and Pleural Indentation represent the malignant lung nodule lesions. Air Bronchogram, Obstructive Pneumonia, and Calcification are classified as benign or malignant lesions. Figure 5 presents the sites of nine lung nodule signs in images in the database.

(ii) LIDC-IDRI Dataset

The LIDC-IDRI dataset is the world’s biggest publicly available database for researchers and consists of thoracic CT images (1018 CT scan images) from 1010 patients (Figure 6). These images show the presence of lung cancers and nodule annotations, which means outlines. The nodule annotations are available for all patients. Still, the diagnostic information is presented for 157 patients, which consists of nodule information ratings, where 0 is an unknown class, 1 is the benign class, 2 is the primary malignant class, and 3 is the metastatic (malignant) class.

The ratings assigned for each image are based on a biopsy, surgical resection, progression, and the revision of radiological images. The diagnosis class is classified into two classes: (i) patient level and (ii) nodule level. To make this dataset for radiologists, images from 1000 patients were obtained over a long time with different CT scanners. We present the diameter size distribution for lung nodules in the LIDC-IDRI database in Table 6. Multiple lung nodules accompany the different slices. Each lung nodule slice comprises annotated data, and its presence is based on a 3 mm size.

In the LIDC database, diagnostic data are the means of ascertaining malignancy. We chose the ratings from diagnostic data; the ground truth was processed for training in the triplet CBMIR system, and it was evaluated by comparing the results with the radiologist-given ratings in the database. This database provides annotations for nodules with sizes from 3 mm to 30 mm shown in the yellow box.

### 5.3. Performance Measures

Equation (14) shows the accuracy rate, which is the number of accurate illness estimates relative to the total number of diagnoses. The capacity of a predictive result to reliably recognize persons without any signs of illness risks in the lung is provided in Equation (15). and represents specificity. The capacity of a decision outcome to properly identify persons with the illness is characterized as sensitivity, which is estimated in Figure 8.

The number of actual positives (i.e., risk controls and indicators for people who belong to the true-positive group) relative to the total number of people identified by the classification model (i.e., the total number classified as positive outcomes, including components incorrectly labeled as belonging to the positive class) equals the precision for a category.

The F-Score is the harmonized average mean of the specificity and sensitivity scores, as shown in Figure 11. The processing time delay is the cumulative time lag in learning the computational model and then evaluating it. It can be seen in Equation (15). In the following, the performance of the lung cancer diagnosis model is discussed and compared to existing methods. Table 7 illustrates the formulas for the performance metrics.

Figure 7, Figure 8, Figure 9, Figure 10 and Figure 11 illustrate the performance of the model based on the accuracy, sensitivity, specificity, precision, and execution time concerning the number of instances. Based on the results shown in the figures, the performance of the proposed ML-CNN method produces the best results for lung cancer diagnosis. As a result of effective data collection, the preprocessing of images, extracted features, selected features, and classification by tuning input parameters, the performance of the proposed ML-CNN method has reached the maximum level. By addressing the limitations of poor-quality images, insufficient data, and features, the performance is improved. Furthermore, the filtering accuracy is computed for the proposed method, and the existing methods support the classification accuracy when the image-based lung cancer diagnosis is the focus.

After the selection of features (visual and semantic), they are subjected to a denoising process to remove noise, if any, present in them. To evaluate the efficiency of the method in this paper, filtering accuracy was analyzed. This filtering accuracy metric defines the range of filtering, i.e., whether the frames with noise are completely removed. The filtering accuracy is formulated in terms of the inverse of the mean square error, which is given as
(14)$$\mathrm{Filtering}\;Accuracy=\left(\frac{1}{MSE}\right)*\;100\;\%$$
where the mean square error is
(15)$$\mathrm{MSE}=\frac{1}{n}\;\sum_{i=1}^{n}\;{\left(\hat{Y_{i}}-Y_{i}\right)}^{2}$$

In Equation (15), $\;{\left(\hat{Y_{i}}-Y_{i}\right)}^{2}$ is the square of errors, and 1/n ∑_(i = i)^n is the mean. According to the defined formulation, the filtering accuracy was determined and shows good results. The proposed Bayesian threshold method in this research work was compared with the previous fast bilateral filter involved in removing noises from images. The comparative results of the filtering accuracy are demonstrated in Figure 12 and Figure 13, from which the better efficiency of the proposed method is confirmed. This result is due to the involvement of a block-matching filter with a Bayesian threshold that removes errors from all parts of the image.

In contrast, a fast bilateral filter focuses on removing noise that is present at the diagonals only. This is the major reason for the reduced filtering accuracy of fast bilateral filters. The graphical comparison shows a gradual increase in filtering accuracy with the increase in time. An increase in this metric also causes an increase in classification accuracy due to the improvement in the clarity of images. Then, an optimization-based comparison was implemented for the proposed and previous approaches. In this work, hyperparameters were tuned using the PSO algorithm, which was compared to the traditional algorithms. Table 8 compares the other optimization methods to the PSO with their merits and demerits.

We analyzed the suggested approach using a variety of PSO-selected attributes to evaluate its overall performance parameters. F1 = 0.6, F2 = 0.5, and h = 1.0 have the maximum degree of precision. Table 9 represents the test results for various parameters. We ran several preliminary trials to find the optimal empirical combination of particle and iteration numbers. We discovered that 2500 particles and 29 iterations produced the final performance results given in Table 11 and Figure 7.

The PSO data, PSO iterations, and characteristics are described in Table 10, Table 11 and Table 12, which aids the model’s ability to forecast results accurately.

## 6. Conclusions

Virtual access to health imaging, including lung cancer computed tomography, is one of the benefits of IoT-based healthcare institutions. IoT nodes have made it easier to leverage data collected by various servers to investigate illness trends. As a result, a cancer diagnosis can be made by utilizing this information for CNN training. We developed a new classifier based on a thorough fully convolutional neural network in this study. ML-CNN is a broad sense classifier that may be used to detect and categorize biological imagery. However, this work uses ML-CNN to detect and categorize pulmonary nodules in CT scan images. The proposed ML-CNN classifier is trained on two categories: nodule (diseased—malignant or normal) and non-nodule (non-diseased—malignant or benign) (normal). Firstly, the sensor information is preprocessed using the normalization technique and then forwarded to the ML-CNN with PSO for feature extraction and classification. Secondly, the lung images are preprocessed using a Taylor series-based Bayesian threshold for noise reduction, and then Z-score normalization is used to normalize the intensity values. Then, the lung-image-based information is extracted from the image, such as color, shape, and intensity, and the performance is improved by tuning the hyperparameters using the PSO approach.

ML-CNN with PSO provides accuracy, precision, sensitivity, specificity, and F-measure values of 98.45, 98.89, 98.45, 98.62, and 98.85, respectively. The hybrid method gives high precision with greater convergence results when compared to other methods.

In deep learning, hyperparameters are parameters whose values are used to control the learning process. If we want to find some hidden aspects of the CNN, such as color, shape, and intensity, better performance is achieved by adjusting the hyperparameters. Model hyperparameters are manually set and used to help estimate model parameters. Model hyperparameters are often called parameters because they are a part of machine learning that must be manually fine-tuned.

PSO improves the accuracy, precision, sensitivity, specificity, and F-measure for both PSO and hyperparameters, which helps the CNN in feature extraction and classification.

We plan to employ more lung datasets in future research and apply hypotheses via fine-tuning and transfer learning to categorize different types of CT images, including melanoma, mammogram, and heart images. The model’s generalization for diverse types of medical imaging will also be evaluated in the continued study.

## Figures and Tables

**Figure 1 bioengineering-10-00320-f001:**
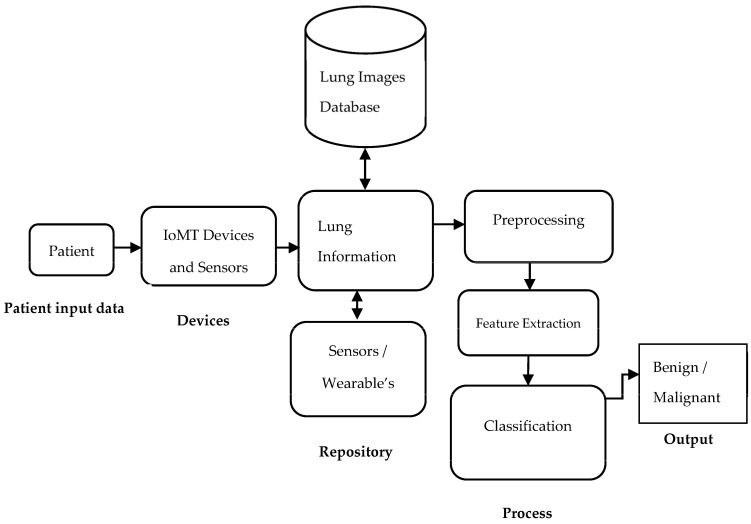
Pipeline for IoMT-based lung disease diagnosis.

**Figure 2 bioengineering-10-00320-f002:**
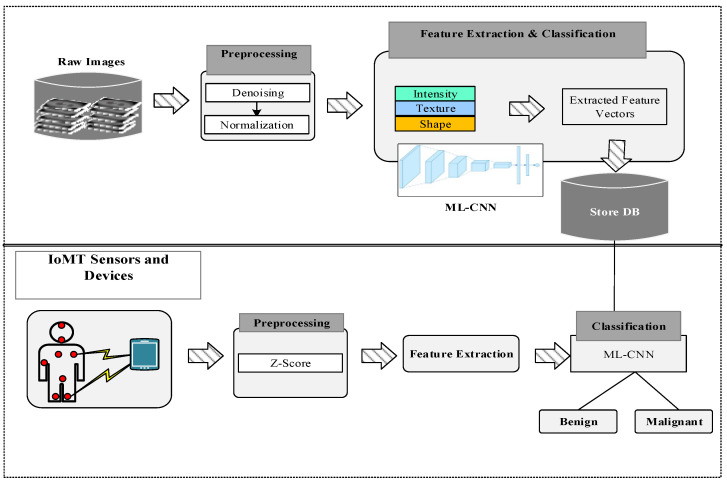
System architecture.

**Figure 3 bioengineering-10-00320-f003:**
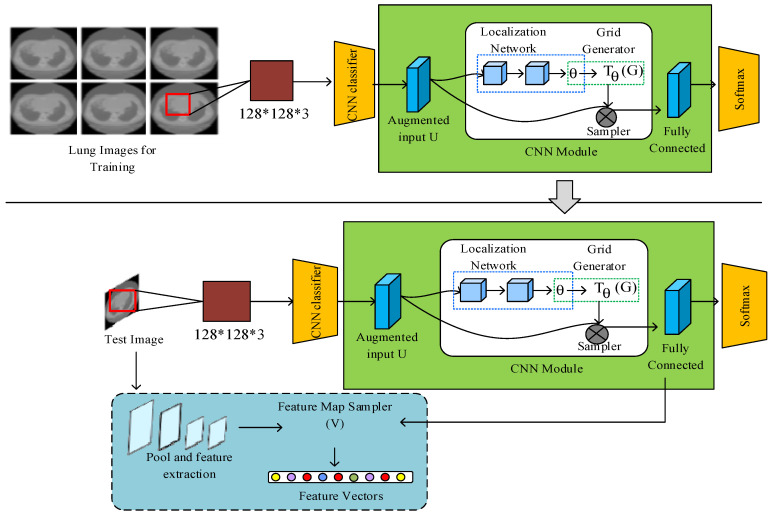
Machine Learning—CNN for classification.

**Figure 4 bioengineering-10-00320-f004:**
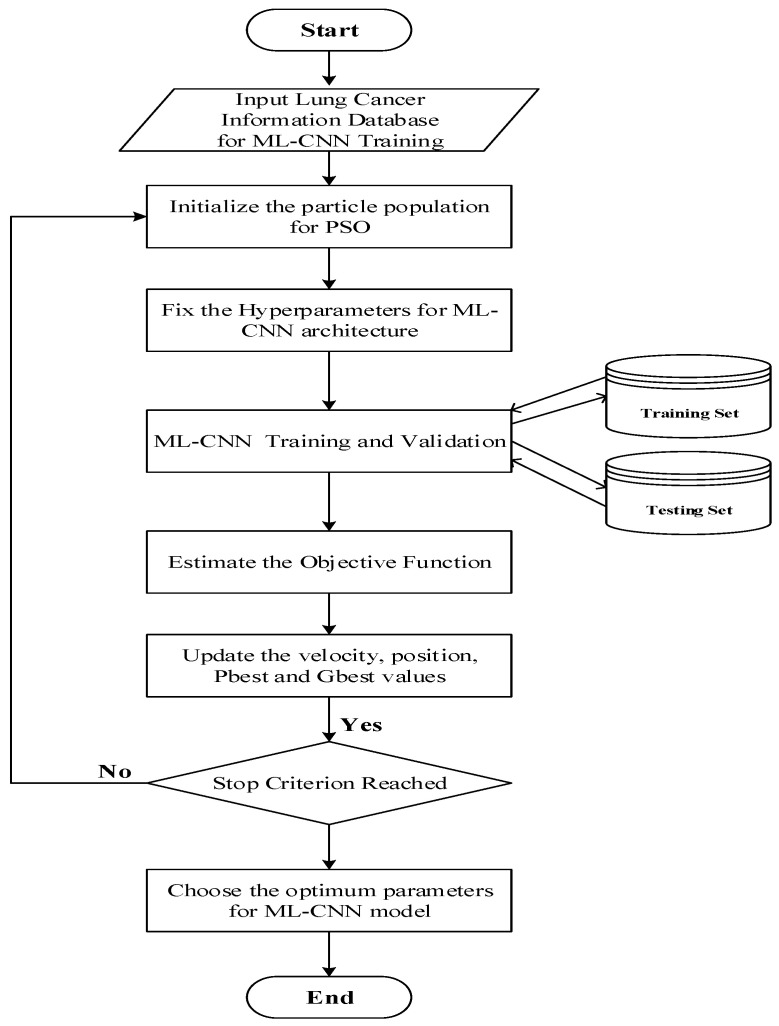
Flowchart for CNN with PSO.

**Figure 5 bioengineering-10-00320-f005:**
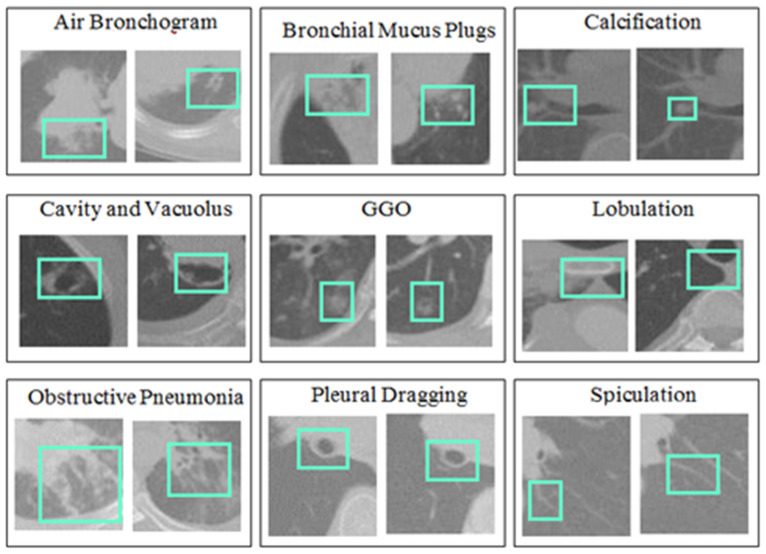
Abnormal signs in LISS dataset.

**Figure 6 bioengineering-10-00320-f006:**
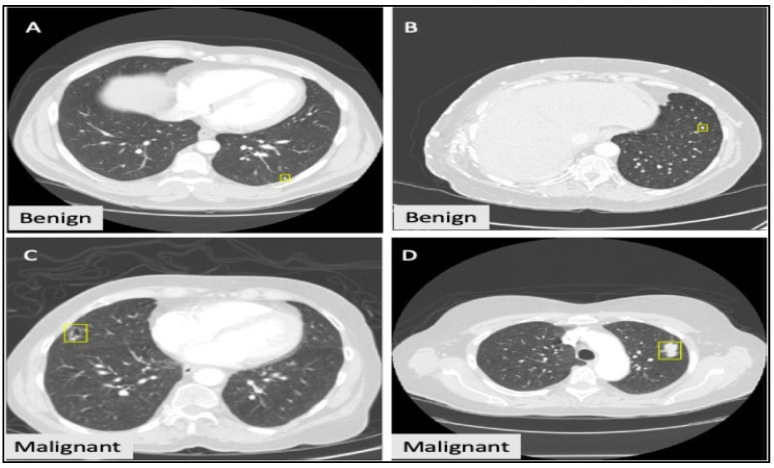
Lung classification results.

**Figure 7 bioengineering-10-00320-f007:**
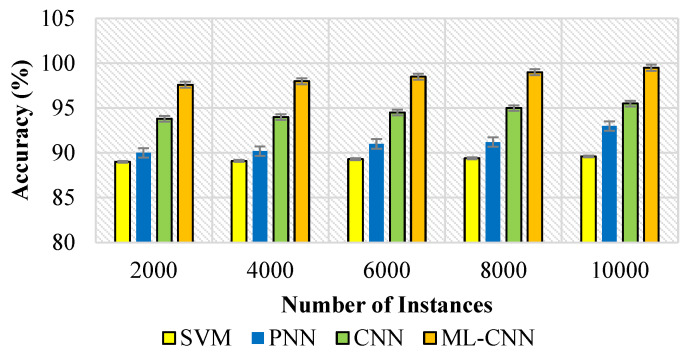
Accuracy vs. number of instances.

**Figure 8 bioengineering-10-00320-f008:**
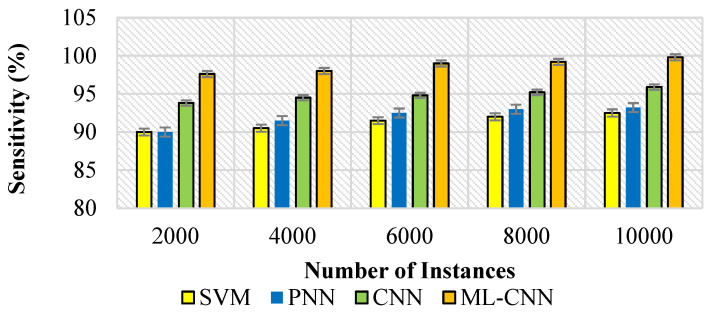
Sensitivity vs. number of instances.

**Figure 9 bioengineering-10-00320-f009:**
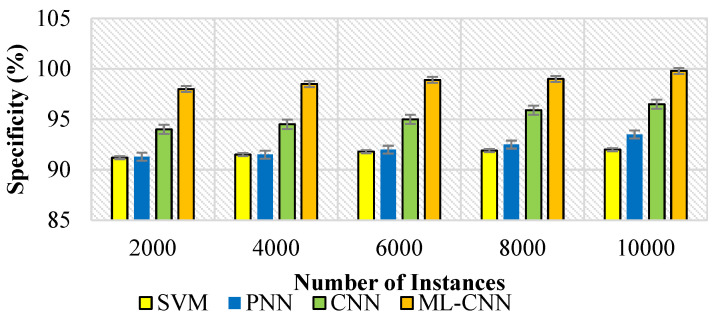
Specificity vs. number of instances.

**Figure 10 bioengineering-10-00320-f010:**
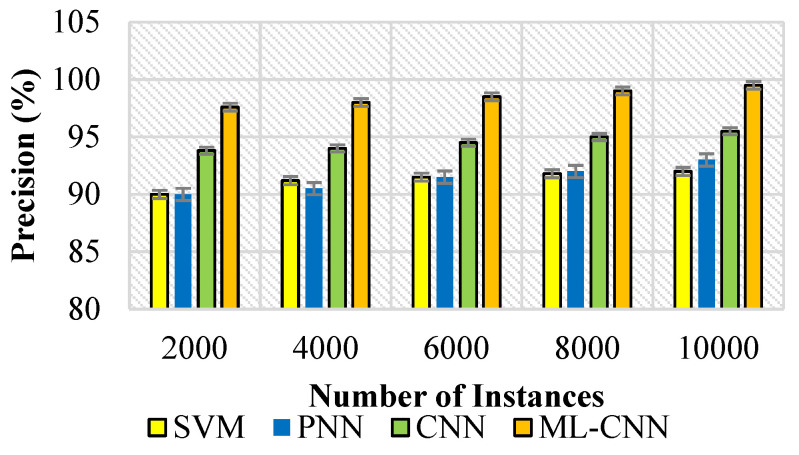
Precision vs. number of instances.

**Figure 11 bioengineering-10-00320-f011:**
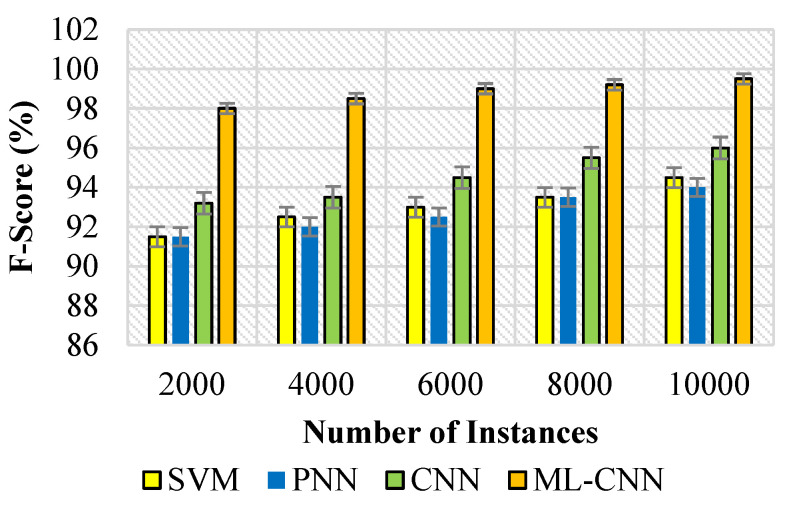
F-score vs. number of instances.

**Figure 12 bioengineering-10-00320-f012:**
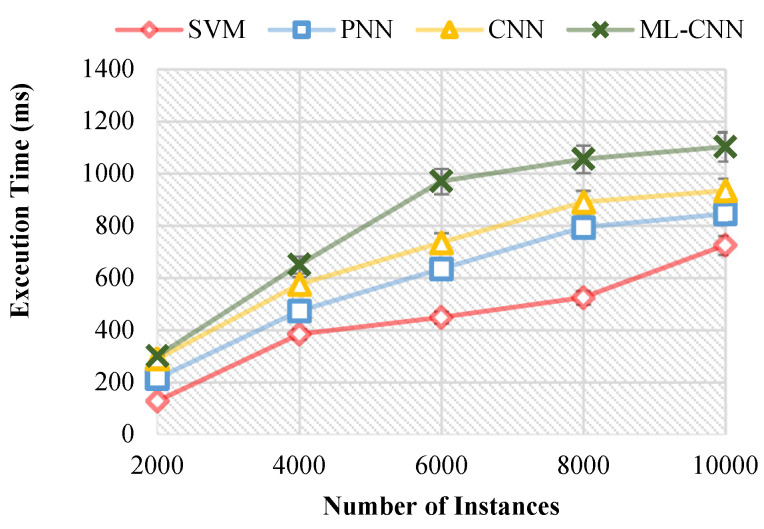
Execution time vs. number of instances.

**Figure 13 bioengineering-10-00320-f013:**
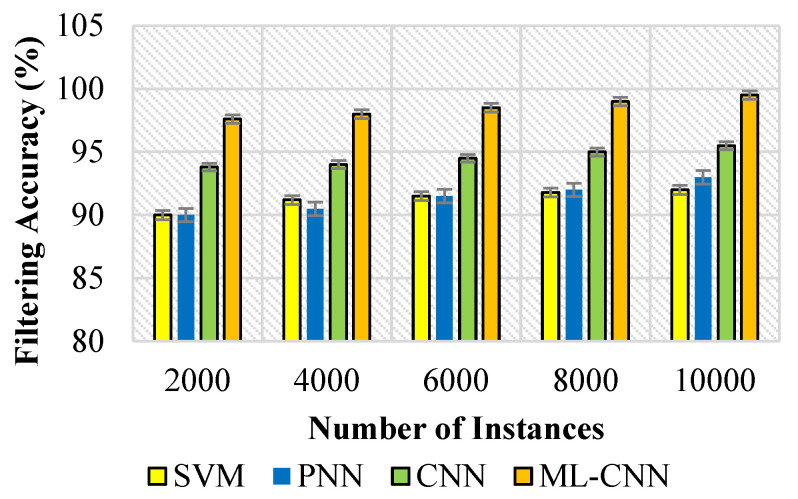
Filtering accuracy vs. number of instances.

**Table 1 bioengineering-10-00320-t001:** List of extracted features.

Feature Type	Feature	Description	Formula
Texture features	Correlation	It is defined by the similar direction of rows and columns	$\overset{n-1}{\underset{i,j=0}{\sum }}p_{i,j}{\left(i-j\right)}^{2}$
	Contrast	It indicates the brightness or contrast for each nodule. When the contrast level is high, texture features are not clear.	$\overset{n-1}{\underset{i,j=0}{\sum }}p_{i,j}\left(\frac{\left(i-\mu_{i}\right)\left(j-\mu_{j}\right)}{\sigma_{i}\sigma_{j}}\right)$
	Homogeneity	It is defined as the distribution closeness of entities in the matrix. It is computed in all directions of the image.	$\underset{i,j}{\sum }\frac{p\left(i,j\right)}{1+\left|i-j\right|}$
	Entropy	It is defined by the complexity of textures or unevenness	$E=-\overset{G}{\underset{g=1}{\sum }}\overset{G}{\underset{g=1}{\sum }}p\left(g,g^{\prime }\right)\log p(g,g^{\prime })$ p(i,j) is the normalized image matrix for one pixel coordinate, n is the total number of distinct gray levels in the image
Shape features	Area	The number of pixels in the largest axial slice computes it $lar_{aS}$ multiplied by the resolution of the pixels	I(x,y)×ΔA
	Aspect Ratio	It is computed by the major axis and minor axis length	$\frac{MA\left(x,y\right)\left[L\right]}{MI\left(x,y\right)\left[L\right]}$
	Roundness	It is computed by the similarity of the lung nodule region to the circular shape	$\frac{4\pi A}{L}$
	Perimeter	It is defined as the structural property for the list of coordinates and also the sum of the distance from each coordinate	$\sqrt{{\left(X_{i}-X_{i-1}\right)}^{2}+{\left(Y_{i}-Y_{i-1}\right)}^{2}}$
	Circularity	It is computed by the 〖lar〗_aS of each lung nodule image	$\frac{4\pi A}{Q^{2}}$ ΔA is the area of one pixel in the shape of I(x,y), $X_{i}$ and $Y_{i}$ is the ith pixel coordinates, A is the nodule area, and L is the nodule region boundary length, MA(x,y)[L] is the major axis length and MI(x,y)[L] is the minor axis length.
Intensity features	Uniformity	It is defined by the intensity uniformity of the Histograms	$\sum_{i=0}^{l-1}H^{2}\left(R_{i}\right)$
	Mean	The average intensity measure represents it	$\sum_{i=0}^{l-1}R_{i}\times H\left(R_{i}\right)$
	Standard Variance	It is represented by the 2nd moment of the average values	$\sum_{i=0}^{l-1}{\left(R_{i}-M\right)}^{3}\times H\left(R_{i}\right)$
	Kurtosis	It is defined by the 4th moment of mean	$\sum_{i=0}^{l-1}{\left(R_{i}-M\right)}^{4}\times H\left(R_{i}\right)$
	Skewness	It is defined by the 3rd moment of the mean values	$\sum_{i=0}^{l-1}{\left(R_{i}-M\right)}^{3}\times H\left(R_{i}\right)$
	Smoothness	The relative intensity differences in the given region define it	$1-\frac{1}{\left(1+\sigma^{2}\right)}$ $R_{i}$ Is a random attribute that indicates the intensity, H(R_i) is the Histogram of the Intensity in the nodule region, l denotes the number of potential intensity levels, and σ represents the standard deviation

**Table 2 bioengineering-10-00320-t002:** List of extracted low-level features.

	Feature Name
Texture features	Correlation
	Contrast
	Homogeneity
	Sum of square variance
	Spectral, Spatial, and Entropy
Shape features	Area
	Irregularity
	Roundness
	Perimeter
	Circularity
Intensity features	Intensity
	Mean
	Standard Variance
	Kurtosis
	Skewness
	Median

**Table 3 bioengineering-10-00320-t003:** High-level (semantic) features for lung nodule classification.

Nodule Size	<4 mm, 4–7 mm, 8–20 mm, and >20 mm
Speculation	The range between 0 and 1
Diameter	Nodule diameter
Location	Upper lobe or not
Morphology	Smooth, Lobulated, Spiculated, and Irregular

**Table 4 bioengineering-10-00320-t004:** Hyperparameters of CNN and PSO.

Methods Used	Parameters	Description
CNN	Number of Epochs	5
	Non-Linear Activation Function	ReLU
	Activation Function	Softmax
	Learning Function	Adam Optimizer
PSO	Number of Particles	10
	Iterations	10
	Inertial Weight (W)	0.85
	Social Constant (W2)	2
	Cognitive Constant (W1)	2

**Table 5 bioengineering-10-00320-t005:** Dataset description.

	CT Imaging Scans	Annotated ROIs
GGO	25–2D 19–3D	45–2D 166–3D
Lobulation	21	41
Calcification	20	47
Cavity and Vacuoles	75	147
Speculation	18	29
Pleural Dragging	26	80
Air Bronchogram	22	23
Bronchial Mucus Plugs	19	81
Obstructive Pneumonia	16	18
**Sum**	271	677

**Table 6 bioengineering-10-00320-t006:** LIDC-IDRI database description.

Nodule Description	Number of Nodules
No. of nodules or lesions marked by at least 1 radiologist	7371
No. of <3 mm nodules marked by at least 1 radiologist	2669
No. of <3 mm nodules marked by all 4 radiologists	928

**Table 7 bioengineering-10-00320-t007:** Parameters for lung cancer diagnosis.

	Formula
Accuracy	$ACC\;=\;\frac{TP\;+\;TN}{TP\;+\;TN\;+\;FP\;+\;FN}\;\times \;100\%\;$
Sensitivity	$Sensitivity\;=\;\frac{TP}{TP\;+\;TN}\;\times \;100\%\;$
Specificity	$Specificity\;=\;\frac{TN}{TN\;+\;FP}\;\times \;100\%\;$
Precision	$Precision\;=\;\frac{TP}{TP\;+\;FP}\;\times \;100\%\;$
F-Score	$F=\frac{2\;*\;Precision\;*\;Sensitivity}{Precision\;+\;Sensitivity}\;\times \;100\%$
where TP = true positive, TN = true negative, FP = false positive, FN = false negative	

**Table 8 bioengineering-10-00320-t008:** Comparison of optimization approaches.

Approach	Description	Merits and Demerits
Genetic algorithm	A genetic algorithm is an evolutionary algorithm that deals with chromosomes. Operations such as crossover and mutation are performed.	Merits: 1. Parallel processing 2. Flexible
		Demerits: 1. Gives close to the optimal solution 2. Lot of computations are required
Ant Colony Optimization	This algorithm is based on the behavior of ants, which deposit chemical components called pheromones.	Merits: 1. Faster discovery with appropriate solutions 2. Solves travel salesman problem
		Demerits: 1. Inappropriate for continuous problems 2. Weakened local search
Bee Colony Optimization	Individual proactive abilities and self-organizing capacity are developed with honey bee behavior	Merits: 1. Faster convergence 2. Uses fewer control parameters
		Demerits: 1. Allows only a limited number of search space
Particle swarm optimization	The population-based algorithm initializes the input and evaluates the fitness value.	Merits:– 1. Robust 2. Less computation time

**Table 9 bioengineering-10-00320-t009:** Comparison of numerical results for proposed and existing methods.

Algorithm	Attacks	Accuracy (%)	Precision (%)	Sensitivity (%)	Specificity (%)	F-Measure (%)
GA	2000	84.52	86.75	85.45	86.75	85.7
	4000	86.54	87.45	86.12	87.45	85.9
	6000	87.45	88.15	86.75	88.78	85.6
	8000	88.02	88.16	86.78	88.95	85.9
	10,000	88.15	88.19	86.79	88.23	86
ACO	2000	88.45	88.78	88.91	88.96	89
	4000	88.6	88.75	88.90	88.97	89.15
	6000	88.69	88.89	88.92	88.98	89.45
	8000	88.72	88.90	88.94	88.98	89.55
	10,000	89.75	88.91	88.95	88.99	89.6
BCO	2000	90.1	91.23	91.25	91.44	91.63
	4000	90.25	91.45	91.35	91.56	91.65
	6000	90.26	91.75	91.39	91.58	91.67
	8000	90.32	91.88	91.40	91.60	91.69
	10,000	90.45	91.89	91.45	91.62	91.85
PSO	2000	98.1	98.23	98.25	98.44	98.63
	4000	98.25	98.45	98.35	98.56	98.65
	6000	98.26	98.75	98.39	98.58	98.67
	8000	98.32	98.88	98.40	98.60	98.69
	10,000	98.45	98.89	98.45	98.62	98.85

**Table 10 bioengineering-10-00320-t010:** Algorithm parameters for PSO using empirical data.

	F2	h	Accuracy
0.8	0.6	1.0	98.45
0.8	0.6	0.9	97.73
0.8	0.6	1.0	98.12
0.7	0.6	1.0	98.09
0.6	0.5	1.0	99.46

**Table 11 bioengineering-10-00320-t011:** PSO method results when utilizing a constant number of particles and an increasing number of iterations.

	Iterations	Accuracy	Precision	F-Measure
2500	25	97.90	97.89	97.12
2500	26	98.06	97.03	97.56
2500	27	98.45	96.43	96.49
2500	28	98.23	97.63	98.62
2500	29	99.56	99.54	99.32
2500	30	97.96	97.87	97.51

**Table 12 bioengineering-10-00320-t012:** Observations of the PSO algorithm with different feature sizes.

Features	Accuracy	Precision	F-Measure
10	99.45	99.03	99.89
12	98.09	97.46	97.43
15	98.83	98.03	98.69
18	98.23	98.67	97.52
20	97.12	97.23	98.86

## Data Availability

On request we can produce the data to the concern person.

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
