# Peer review of "Optimization System Based on Convolutional Neural Network and Internet of Medical Things for Early Diagnosis of Lung Cancer"

_bioengineering, 2023, doi:10.3390/bioengineering10030320_

Round 1

Reviewer 1 Report

This paper proposed an optimization system using ML-CNN and internet of medical things for early diagnosis of lung cancer. The paper is meaningful and interesting, while some issues should be further clarified as follows:

Q1: The motivation of this paper is unclear, what are the research gaps of early diagnosis of lung cancer?

Q2: The contributions of this paper are suggested to highlight. Particularly, the CNN and internet of medical things are regular technologies in the field of computer vision.

Q3: Actually, the comparative analysis uses genetic algorithm, ant colony optimization, bee colony optimization, and particle swarm optimization, which is unfair and unreasonable. These competitors are all machine learning methods, they are usually inferior to the deep learning methods.

Q4: If possible, the necessary ablation experiments should be provided in this paper.

Q5: What is the bench line of this method? Authors are suggested to provide the bench line in this paper.

Q6: There are many spelling and grammatical mistakes in this paper, authors are suggested to polish and revise the entire paper.

Reviewer 2 Report

The manuscript entitled "Optimization System Based on CNN and Internet of Medical Things for Early Diagnosis of Lung Cancer” has been investigated in detail. The topic addressed in the manuscript is potentially interesting and the manuscript contains some practical meanings, however, there are some issues which should be addressed by the authors:

·         Why the features of frequency and time-frequency domain were not used?

·         Which kernel has been used for the support vector machine? It should be mentioned in the manuscript.

·         The resolution of the figures should be improved.

·         I recommend the authors to review other recently developed works.

·         Please add numerical result to the conclusion.

·         I suggest engaging in professional proofreading to restructure the paper language.

This study may be consider for publication if it is addressed in the specified problems.

Reviewer 3 Report

Comments:

The manuscript entitled “Optimization System Based on CNN and Internet of Medical Things for Early Diagnosis of Lung Cancer” by Yossra and et al, in which the prediction of lung cancer was realized by using ML-CNN in deep learning and Internet of Things. Moreover, compared with the existing support vector machine (SVM), probabilistic neural network (PNN), traditional CNN and other methods, it has advantages in accuracy, precision, sensitivity, specificity, F-score, calculation time and other aspects, which can help radiologists to diagnose early lung cancer, and has a good application prospect. I believe that this paper can be accepted after addressing these following issues:

1. References 1-20 cited were not found in the text.

2. There are many formatting errors in the text:

(1) In Abstract: “Compared to other techniques, These experimental results…”

(2) In Abstract: “The performance of the proposed ML-CNN is employed using Python, where accu-racy (2.5%–10.5%) is high when compared to the number of instances, precision is (2.3–9.5%) high…”  The format of all percentage intervals in the text should be consistent.

(3) Whether “AI” is short for “Intelligent Systems” or “Artificial intelligence”, the expression is inconsistent in the paper, so it needs to be more clear.

(4) “Consideration of IoT sensor and medical images: …” The first letter should be capitalized, consistent with the previous text.

(5) “Lastly, depending on the many available ap-proaches, methods have been defined and depicted.” Word repetition.

(6) “The unclear features of the lung cancer nodule make computer-aided automated diagnosis difficult at the moment. Average ac-curacy.” “Average ac-curacy.” The expression here is not clear, appears somewhat abrupt.

(7) “TCM has been proven to be an excellent therapy for treating lung disease,” “TCM” appears for the first time in the text and should be given its full name.

(8) “?? is the thresholding obtained for all bands?” “?” should not appear.

(9) The formula is not numbered.

(10) “Equation 7 ??? (??,??+?) denotes the distance measure of relative entropy and equa-tion 5.3 7 ??? (??,??+?) Gives the distance measure of and Square Root of Relative En-tropy between the frames.” Unclear expression.

(11) “?? Is a random attribute…”, “Where ??(?+?) Represents the…”

(12) Please use the same font format for all the pictures in the article.

(13) The format of references needs to be standardized and uniform.

3. “6.1. Preprocessing” and “6.2. Preprocessing” contain the same statements.

4. Figure 1 requires a clearer substitution.

5. The illustration should be annotated.

6. I also suggest the authors add “Author Contributions, Institutional Review Board Statement” in the manuscript.

7. Personally, I think the content of “2. Primary Knowledge”, “3. Related Work”, “4. Motivation for lung cancer research” and “5. Problem Statement” can be clearly elaborated in “Introduction”.

Reviewer 4 Report

The authors present a structured approach to lung cancer diagnosis with machine learning which is based on an Internet of Things framework and context. But the authors need to more clearly explain in the Introduction and Discussion/Conclusion what is the contribution of their work, since CNN for lung cancer diagnosis is not itself novel of course (as they describe in detail in Related Work). It appears that the novelty is in the optimization method for hyperparameter tuning, but most of the Introduction and background section deals with the aspects of the CNN -- which may not be novel? It is not clear. If the contribution is the comparison of hyperparameter optimization methods and conclusion that particle swarm is better than other approaches, the paper should be rewritten and edited to better focus on that.

The paper is largely OK in writing, but it requires a thorough editorial review. For example towards the bottom of pg. 4 there is a sentence " Conclude: The approach for selecting relevant feature extraction is used to increase classification performance" This doesn't make sense and seems to be from a rough draft. It can be deleted.

The "Related Work" section is very comprehensive and a strength of the paper. But because of its length it needs more structuring. The very long paragraph should be broken up. It is OK to have short paragraphs, even just 1 per reference or group of related references. A paragraph may even be a single sentence if it is easier for a reader to step through. Also, because the authors are using citation numbers to label studies, which is ordinarily fine, it is hard to track what they are doing chronologically or by topic. So some sense of chronology (year of study, e.g.) and signposting, i.e., labeling of the category "Studies using X method..." etc. as an initial sentence, would be helpful in this section. Right now it is hard to read and properly evaluate. Also, unless I'm missing something, the paper says that "65 articles" were reviewed, but I don't see a description of those articles. Also, the disadvantages are not really summarized or explained for the related work articles, despite the authors purporting that they would explain them in stepwise fashion.

Section 4 "Motivation" and section 5 "Problem Statement" can be deleted. They are not contributing anything to this work. Rather, what is needed for the introduction is a motivation for why the *specific* approach described in *this* manuscript. What is the need? If the related work are already performing well, what is the goal of this work? What are existing methods failing to do? The authors should describe this with specificity. Sections 4 and 5 are just generalities. 

There appears to be a typo since the content of 6.1 is repeated at the beginning of 6.2. 

There is a lot of material about the CNN architecture, but not clearly how the hyperparameter optimization methods were evaluated -- how those experiments were structured to avoid overfitting, whether an external validation set was used, and how the choices of parameters for the optimization methods was made -- since as heuristic methods, their hyperparameters matter too. If this is the point of novelty in the paper, it needs to be rewritten so that we can properly evaluate this in peer review.

Similarly, the Discussion/Conclusion should attempt to explain why PSO might be better than other methods. Right now one might suspect it is just because the other optimization methods had relatively poorer parameter choices.

In sum, I respectfully suggest major revisions to the text to better focus on the point of novelty in the study. While I have graded the paper low for Novelty and Significance of Content, that may be improved simply with clearer explanation of these things.

Round 2

Reviewer 1 Report

Actually. this paper is a fairly straightforward application/modification of existing algorithms(e.g., CNN and PSO). The distinctive contributions to advanced medical imaging are weak. 

Reviewer 2 Report

My recommendation is "Accept".

Reviewer 4 Report

The authors have revised the manuscript to address many of the substantive concerns previously. There is still a need for some English editing: Just as two selected examples, there is first person ("I" statements) in Lines 106-110 when there are multiple authors; or at line 679 "Computed" is capitalized. I would suggest a thorough editorial review prior to publication.
